# Role of Vitronectin and Its Receptors in Neuronal Function and Neurodegenerative Diseases

**DOI:** 10.3390/ijms232012387

**Published:** 2022-10-16

**Authors:** Yelizhati Ruzha, Junjun Ni, Zhenzhen Quan, Hui Li, Hong Qing

**Affiliations:** School of Life Science, Beijing Institute of Technology, Beijing 100081, China

**Keywords:** vitronectin, neuron, neurodegenerative diseases, blood–brain barrier

## Abstract

Vitronectin (VTN), a multifunctional glycoprotein with various physiological functions, exists in plasma and the extracellular matrix. It is known to be involved in the cell attachment, spreading and migration through binding to the integrin receptor, mainly via the RGD sequence. VTN is also widely used in the maintenance and expansion of pluripotent stem cells, but its effects go beyond that. Recent evidence shows more functions of VTN in the nervous system as it participates in neural differentiation, neuronutrition and neurogenesis, as well as in regulating axon size, supporting and guiding neurite extension. Furthermore, VTN was proved to play a key role in protecting the brain as it can reduce the permeability of the blood–brain barrier by interacting with integrin receptors in vascular endothelial cells. Moreover, evidence suggests that VTN is associated with neurodegenerative diseases, such as Alzheimer’s disease, but its function has not been fully understood. This review summarizes the functions of VTN and its receptors in neurons and describes the role of VTN in the blood–brain barrier and neurodegenerative diseases.

## 1. Introduction

The extracellular matrix (ECM) is a protein network system composed of various macromolecules around cells, which provides mechanical support and physical strength to cells and plays an important role in cell adhesion, migration, proliferation and differentiation [1]. In the central nervous system (CNS), the ECM can bind all kinds of neurons and glial cells together to form the basic structure of the CNS. The ECM is also the external environment that neurons depend on for survival, which has the functions of nourishing and protecting neurons, and promoting neuronal synapse growth and stem cell differentiation. The main components of the ECM include collagen, laminin, vitronectin, fibronectin, etc. [2].

Vitronectin (VTN), also known as S-protein or serum diffusion factor [3], is an ECM protein that acts as the master controller of the extracellular environment [4]. It can promote endothelial cell adhesion and plays a key role in tissue remodeling [5]. Moreover, VTN participates in multi-protective events through integrins. As the receptor of VTN, integrin is the main way for cells to bind and respond to VTN. Integrins αvβ3 and αvβ5 have been found to play important roles in enhancing VTN function, promoting blood vessel formation in mice [6] and early hematopoietic differentiation of human pluripotent stem cells [7]. VTN and its receptors have been linked to several diseases, including tumors, coagulation disorders and inflammatory diseases. Recently, VTN was reported to play an essential role in regulating neuronal activities and functions, while there is still a lack of research on its role in neurological diseases. In this review, we summarize the current knowledge and findings regarding VTN in the area of neuron activities and neurodegenerative diseases. We then review the interactions between VTN and its receptors and the blood–brain barrier. After that, we display the current targeted drugs for VTN and its receptors. Finally, we briefly discuss the future perspectives of VTN.

## 2. Function and Structure of VTN and Its Receptor

VTN can be found in plasma [5] and platelets [8]. It distributed in the human liver [5] and spleen [9], and also in the atrioventricular cushions of embryonic chick hearts [10]. VTN was originally found in Hela and it is essential for conjunctival and cardiac cell growth. Hela, conjunctiva and human heart cells cease to grow if VTN is removed [11]. In humans, VTN is located on chromosome 17Q11.2 and consists of eight exons and seven introns [12]. With a length of 478AA and a size of 54306Da, the protein exists in two forms: the single-stranded 75-kDa form (V75) or the two-stranded form (65 kDa and 10 kDa) [13]. VTN is a conformational molecule and this conformational variability may have a large impact on its functional changes. It can promote the extension and proliferation of cells and can be used in the study of cell migration experiments [5]. Among many biological effects, it has been found that VTN regulates neuronal functions including cell differentiation, neuroprotection, neurogenesis, etc. (Figure 1). For example, VTN was demonstrated to promote the differentiation of human embryonic stem cells [14], oligodendrocytes [15] and mouse cerebellar granule cell precursors [16]. Moreover, VTN with αvβ3 and αvβ5 integrins is involved in the morphological transition of neurites in the neurogenesis of neuroblastoma cell line neuro2a [17]. Recent evidence, however, suggests a broader role for VTN in neurological diseases. Misfolding of this protein may lead to age-related diseases. Studies showed that, in vitro, VTN inhibits β-amyloid aggregation to form a fibrous amyloid [18], which is the main feature of Alzheimer’s disease (AD). At the same time, dysregulated adhesions of VTN may increase risk mutations associated with AD [19].

VTN is mainly divided into the SMB domain, hemopexin rich region (HPX), cell attachment site and heparin binding domain. Among them, amino acid residues 45-47 are ARG-Gly-ASP (RGD) as the anchor site of the integrin receptor [20]. Instead of making direct contact with cells, VTN usually interacts with cells through integrins in the cell membrane. Integrins belong to a family of cell adhesion receptors that link the ECM to the cytoskeleton. Integrin consists of α and β subunits, and its crystal structure is divided into three parts: the extracellular domain, transmembrane region and cytoplasmic tail. Regulation of VTN by binding to appropriate integrins can enhance inactivating potassium current (IA) amplitude in developing hippocampal neurons and appears to be selective for IA [21]. The extracellular domain of integrins can specifically bind to VTN to transmit biphasic signals [22], mainly mediated by the RGD sequence [23], and VTN binds to specific cell surface receptors such as αvβ3 and αvβ5 integrins [7]. αvβ3 is an important member of the integrin family, which consists of a 125 KDa αV (CD51) subunit and a 105 KDa β3 (CD61) subunit. The αvβ3 activation signal pathway consists of two forms, “inside-out” and “outside-in”. “Inside-out” is the binding of the integrin tail to the receptor in the cytosol to control extracellular ligand binding activity [24]. The “outside-in” signal can phosphorylate downstream kinases, such as mitogen-activated protein kinases (MAPK) and phosphoinositide 3-kinase (PI3K) [25]. The other integrin αvβ5, as an endocytic receptor, binds to VTN to participate in its endocytosis and regulate vascular permeability and the barrier function [26]. However, the intensive biological functions of VTN and its receptors still need further study.

## 3. Functions of VTN in Neurons and Glial Cells

### 3.1. VTN and Neural Differentiation

#### 3.1.1. VTN and Embryonic Stem Cells

In the past, we have known that the vast majority of newborn neurons have begun to differentiate and migrate before the human fetus is born, but little is known about these key neurodevelopmental events. Embryonic stem cells can proliferate in vitro and remain undifferentiated. Under certain conditions, embryonic stem cells can differentiate in multiple directions and generate multiple functional cells [27]. Changing the ECM and selecting VTN or matrigel can induce different substates of human embryonic stem cell-9 line (H9-ESC) [28]. In vivo, VTN is expressed in the cortex and spinal meninges during development [29]. On the other hand, VTN is expressed in the ventral region of the neural tube and promotes the differentiation of motor neurons [30]. The differentiation of motoneurons is enhanced by the synergistic interaction of N-SHH and VTN, and VTN may be required for the correct presentation of morphogen N-SHH to one of its target cells to differentiate motoneurons. In vitro, PVDF nanofiber scaffolds coated with VTN can promote neural differentiation of human embryonic stem cells [14].

#### 3.1.2. VTN and Cerebellar Granule Cells

In addition to embryonic stem cells, VTN also acts on granule cells. Granule cells are the only excitatory neurons in the cerebellum, which use glutamate as a neurotransmitter and express receptors of both subtypes a and b [31]. It has been reported that VTN can promote the progression of the initial differentiation stage of cerebellar granule cells [32], and the signal to exit the precursor cycle of granule cells may come from VTN [33]. Meanwhile, phosphorylation of cyclic-AMP responsive element-binding protein (CREB) induced by VTN regulates the differentiation of granule neurons [34]. In addition to the regulation of CREB phosphorylation, VTN can also regulate the axon specification of mouse cerebellar granule cell precursors through the β5 integrin/PI3K/GSK3β pathway at the differentiation stage [35] and utilize the RGD site to interact with cerebellar granule cells to extend neurites [36]. In particular, VTN was detected to be expressed in human cerebellar Purkinje cells [37], which can produce Shh, and Shh controls the initial proliferation of cerebellar granule cells. Moreover, Purkinje cells are involved in controlling the differentiation time of the cerebellar oligodendrocyte through developmentally regulating the expression of VTN [38].

### 3.2. VTN and Axons

The ECM is a major factor involved in neurogenesis. As an ECM protein, VTN is involved in neural differentiation and plays a role in axonal extension. Axonal growth and orientation are important biological processes, but the underlying cellular mechanisms have not been well explained. Although considerable evidence shows that axon growth and orientation depend on the dynamics of the well-organized cytoskeleton, scientists are still searching for the molecular phenomena that are directly responsible. Axon growth is not only affected by the cytoskeleton, but also requires growth cones to guide axon extension through amoeba movement, which is conducive to axon growth, pathway selection and target cell recognition. In addition, axonal growth is influenced by the ECM, cell-adhesion molecules and their surrounding soluble substances that can enhance and attract or inhibit and repel growth cone growth. Studies have shown that VTN supports the growth of hippocampal neurites and the branching of mouse cortical neurons in vitro [39]. In addition, VTN can interact with the prion protein and its secreted protein stress inducer protein 1 (STI1) to participate in neurogenesis, neuroprotection and memory consolidation [40]. The interaction between cellular prion proteins and VTN supports axonal growth and is compensated by integrins [41]. In particular, VTN has been used as a material for artificial nerve guidance conduits to support and guide neurite extension [42].

### 3.3. VTN and Glial Cells

There is increasing evidence that VTN is not limited to neurons but also closely related to glial cells (Figure 2). Glial cells are widely distributed in the central and peripheral nervous system. They are another large group of cells in addition to neurons, with about ten times the number of neurons, and have the role of supporting and nourishing neurons.

#### 3.3.1. VTN and Microglia

Microglia (MG) are macrophages in the central nervous system, which can be divided into M1 and M2. Under normal circumstances, M1 and M2 maintain dynamic balance. Under the action of IL-13, IL-10, TGF-β and other factors, M1 is polarized to M2, which plays an antiinflammatory role in the later stage of inflammation and promotes damage repair [43]. Neurons degenerate during infection, trauma and ischemia. MG, as a macrophage, will phagocytize and clear damaged and dead neurons [44]. This phagocytosis is the main cause of nerve cell death in animal models such as neuroinflammation and stroke. Cyclic RGD (CRGD), a VTN receptor antagonist, has been reported to inhibit the loss of β-amyloid neurons by blocking the phagocytosis of microglia [45]. CRGD can also improve the morphological and functional parameters of retinal photoreceptor degeneration [46]. The effect of VTN on microglia is also reflected in its interaction with plasminogen activator inhibitor-1 (PAI-1). PAI-1 is a member of the serine protease inhibitor protein family and exists in a variety of conformations, with the secondary “active” form showing inhibitory effects on tissue plasminogen activator (tPA) and urokinase (uPA). Glial cell-derived PAI-1 promotes microglial migration and inhibits phagocytosis by interacting with VTN [47]. In addition, VTN has been reported to be related with TREM2 in patients with Alzheimer’s disease, which is an important gene for maintaining MG metabolism. TREM2 has AD-related mutations R47H, and TREM2 KO has dysregulated adhesion to VTN [19].

#### 3.3.2. VTN and Astrocytes

Astrocytes are the most abundant and widely distributed cell type in the brain [48]. Astrocytes are involved in the generation and maintenance of the blood–brain barrier [49] and secrete trophic factors to regulate synapse formation [50] and neuronal survival [51]. Studies have shown that the ECM is a factor affecting the heterogeneity of astrocyte responses. Primary astrocyte mechanical wound regeneration is dependent on VTN [52]. In addition, VTN interacts with astrocytes in the pericytes of the brain [53]. It has been reported that the VTN-mediated integrin FAK channel acts as a sensor, and FAK can achieve a neurotrophic response by reducing the expression of ciliary neurotrophic factor (CNTF) [54]. VTN released from blood binds integrin on astrocytes to activate downstream FAK signaling through phosphorylation in vitro, enhancing the expression of CNTF to promote the subventricular zone (SVZ) neurogenesis [52]. These studies suggest that VTN-integrin-FAK signaling and the neuroprotective properties of endogenous CNTF may provide novel targets for inducing neurogenesis.

#### 3.3.3. VTN and Oligodendrocytes

Oligodendrocytes are a class of glial cells differentiated from oligodendrocyte progenitor cells (OPCs) and the myelin sheath of the central nervous system is mainly composed of oligodendrocytes [55]. Oligodendrocytes are involved in normal brain development and the repair of demyelinating lesions. Their development process is relatively complex and regulated by many factors [56]. The number of oligodendrocytes in the CNS mainly depends on the migration, proliferation and differentiation of OPCs [57]. It has been reported that the proliferation of OPCs is affected by the integrin signaling pathway. By activating αvβ3 integrin, VTN can help platelet-derived growth factor (PDGF) to promote the OPCs’ proliferation [58]. Regarding oligodendrocyte differentiation, previous studies have shown that VTN, Shh, RA and Noggin cooperate strongly to promote the differentiation of oligodendrocytes [59]. In addition, a novel VTN-derived peptide, VNP2, was recently used to effectively differentiate oligodendrocytes from hPSC-derived OPCs by enhancing SOX10 expression in OPCs [15].

## 4. VTN in Neurodegenerative Diseases

### 4.1. VTN and Age-Related Macular Degeneration

VTN is a major component of abnormal deposits associated with age-related macular degeneration (AMD), Alzheimer’s disease and other age-related diseases [60]. There is increasing evidence that VTN plays a role in the pathologies of neurodegenerative diseases (Table 1). AMD is an age-related neurodegenerative disease, affecting about 170 million people worldwide [61]. Although the pathogenesis of AMD has not been fully revealed, it is currently believed that AMD is a disease caused by the degeneration of retinal pigment epithelial cells (RPE) [62] and photoreceptors [61], leading to severe visual impairment and even rapid blindness. The phagocytosis and digestion ability of RPE on the extracellular segment of retinal photoreceptors becomes weak, resulting in the deposition of undigested residual substances and abnormal metabolites on Bruch’s membrane and the formation of drusen [63]. Previous studies have shown that VTN is a major component of drusen [64] and is expressed in human retinal pigment epithelial cells [65], suggesting that VTN is involved in the pathogenesis of AMD. VTN not only accumulates in drusen, but also interacts with PRE extracellular sediments. Deposits of AMD have rich hydroxyapatite (HAP). A recent study showed that VTN specifically binds soluble calcium ions and the HX domain of solid HAP, and soluble calcium ions can enhance the precipitation of VTN and HAP [60]. In addition, Rs704, an AMD-related variant in the VTN gene, is relevant in the pathogenesis of AMD. Rs704 not only significantly changes the expression, secretion and processing of VTN, but also affects the ability of VTN to bind to retina and endothelial cells [66]. Moreover, VTN was positively correlated with PAI-1 gene expression. Rs704 modification affects the ability of VTN to bind PAI-1 and may contribute to AMD-related vascular changes [67].

### 4.2. VTN and Alzheimer’s Disease

Alzheimer’s disease (AD) is a degenerative neurological disease characterized by memory, cognitive, behavioral and emotional disorders. It currently affects more than 50 million people worldwide and is projected to affect 150 million people by 2050 [68]. The pathogenesis of AD has not been fully elucidated, and there is no effective cure for AD up to now. The main pathological features of AD are senile plaques formed by extracellular β-amyloid deposition and neurofibrillary tangles formed by excessive phosphorylation of the intracellular tau protein [69]. VTN is closely related to Alzheimer’s disease, while it has been reported to exist in senile plaques and neurofibrillary tangles in the entorhinal cortex of AD [70], and its receptor is present in glial cells of senile plaques [71]. Evidence shows that VTN can inhibit the aggregation of β-peptide (Aβ1-42) to form the fibrous amyloid protein in vitro [72]. Moreover, previous studies have shown that VTN and other ECM molecules can regulate the β-amyloid precursor protein [18], indicating there may be a local increase in β-amyloid precipitation. In addition, misfolding of VTN is prone to amyloidosis, which tends to form amyloid [73].

### 4.3. VTN and Multiple Sclerosis

Multiple sclerosis (MS) is a neurodegenerative disease in which the myelin sheath that protects nerve fibers is attacked by the immune system, generating oxidative stress that damages neurons and disrupts the flow of information between the brain and the body [74]. The course of MS involves the death of oligodendrocytes. There is increasing evidence that ECM proteins regulate oligodendrocyte proliferation, survival and development [75]. As discussed above, oligodendrocyte differentiation is influenced by VTN and integrin. Moreover, VTN on astrocytes can regulate the proliferation of oligodendrocyte progenitors, which can act as a link between neuroinflammation and neurodegeneration in MS [76]. VTN is specifically elevated in the vessel wall of active MS and expressed in its plaque proteome [77]. Furthermore, VTN is expressed on short helical axons damaged by active MS and its receptor is expressed on macrophages at the edge of acute lesions and plaques, where myelin damage is most severe [78].

### 4.4. VTN and Stroke

Stroke is an acute neurodegenerative disease that leads to long-term severe motor loss, resulting in impairment of daily activities and quality of life [79]. The pathological mechanism is cerebral ischemia and hypoxic necrosis caused by the sudden rupture or blockage of cerebral vessels, including hemorrhagic stroke and ischemic stroke. After the onset of stroke, the blood–brain barrier is damaged, brain tissue edema is obvious and neurons undergo apoptosis. Interestingly, VTN in mouse stroke models presents gender duality. VTN in female mice only after ischemic stroke, flowed from the blood to the damaged site and SVZ, induced IL-6 increasing through FAK to inhibit post-stroke neurogenesis [80] and aggravated neurological dysfunction [81]. This finding may serve as a target for the detection of stroke in women and offer a new approach to promote neuroprotection after stroke in women. Small vessel disease (SVD) is often named as little stroke. SVD affects arterioles, capillaries and venules that supply white matter and deep structures in the brain [82]. This disease is insidious and slow, and, although the disease is small, it is harmful. In the sporadic small vessel disease (SVD) mouse model, increased VTN expression in endothelial cells affects oligodendrocytes and this interaction may exacerbate white matter damage in SVD [83].

**Table 1 ijms-23-12387-t001:** Overview of VTN in neurodegenerative diseases.

Diseases	Roles of VTN in Neurodegenerative Diseases	References
Age-related macular degeneration	a major component of drusen; involved in the pathogenesis of AMD	[65,66]
Alzheimer’s disease	exists in plaques and neurofibrillary tangles; inhibits the aggregation of β-peptide; regulates β-amyloid precursor protein; misfolding of VTN tends to amyloidosis	[18,70,72,73]
Multiple sclerosis	increased expression in active MS; regulates oligodendrocyte proliferation, survival and development	[75,76,77,78]
Stroke	inhibits post-stroke neurogenesis; aggravates neurological dysfunction	[80,81]

## 5. VTN and Blood–Brain Barrier

The CNS requires an optimal and tightly regulated microenvironment for efficient synaptic transmission. This is achieved by regulating the flux of substances to maintain tissue homeostasis through the blood–CNS barrier [84]. The blood–brain barrier (BBB) is a special structural and functional barrier between the blood circulation system and the nervous system of animals, which is composed of brain microvascular endothelial cells, transporters, the extracellular matrix and its integrin receptors, astrocytes and pericytes. The BBB can express a variety of ion transporters, provide nutrients to the brain through ion channels and remove metabolic wastes in the brain [85,86]. However, not all components can pass through the BBB, which has a highly selective barrier function to limit the passage of pathogens, hydrophilic macromolecules and other substances. By what mechanism does BBB regulate its selectivity and maintain its integrity? There are still many mysteries to be explored about the blood–brain barrier. The ECM is an important part of the BBB. The capillary endothelial cells (EC) that constitute the BBB are tightly wrapped by pericytes (PC). These cells are embedded in the basement membrane (BM) and encapsulated by astrocytes [87]. Recent studies have changed the way we think about VTN, which interacts with integrin receptors to protect the BBB (Figure 3).

### 5.1. VTN Receptor αvβ3 Integrin and Blood–Brain Barrier

The initial stage of the BBB requires the generation of cerebral blood vessels and depends on the adhesion interaction of vascular cells. The VTN receptor is a marker of angiogenesis, among which alpha V beta 3(αvβ3) integrin is a receptor for a variety of ligands in vascular cells [88]. LM609, the antibody of αvβ3, can significantly inhibit tumor-induced angiogenesis in embryonic chick allantoic membrane [89], demonstrating the functional importance of αvβ3 integrin for angiogenesis. Selective ablation of αv integrin in the CNS leads to intracerebral hemorrhage, seizures, axonal degeneration and premature death [90]. Studies have shown that integrins can activate the vascular endothelial growth factor (VEGF) signaling pathway by regulating multiple pathways to affect angiogenesis. The VEGF/VEGFR system activates key integrins in angiogenesis, enhances the recognition and binding of VTN by αvβ3 and promotes cell adhesion, which is evident at low VEGF concentrations [91]. VEGF also enhances integrin αvβ3-mediated cell migration. The need for these integrins in vivo is confirmed by the fact that not only αvβ3 integrins have functions in the angiogenesis, but also that αv integrin knockdown can cause embryonic lethality [92].

### 5.2. VTN Receptor α5 Integrin and Blood–Brain Barrier

VTN is an important ECM that plays an important role in the BBB through its binding to the receptor α5 integrin chain. A recent study showed that in addition to its favorable angiogenic effects, integrin α5β1 has neuroprotective effects to increase the integrity of the BBB [93]. In another study, the regulatory roles of α5 integrin and VTN in the BBB were described in more details. VTN secreted by the CNS pericyte interacts with receptor α5 integrin in vascular endothelial cells through the signaling pathway, which instructs the endothelial cells of the BBB to maintain membrane tension and inhibit transendocytosis, thereby reducing the permeability of the barrier, ensuring that molecules outside the barrier cannot easily pass through, and protecting the safety of the brain [94]. These studies demonstrate that the relationship between VTN and its integrin receptor is critical for barrier integrity and may provide new therapeutic opportunities for CNS drug delivery.

## 6. Current Targeted Drugs for VTN and Its Receptors

The drugs targeting VTN and its receptors are divided into integrin receptor antagonists and platelet receptor antagonists. The former target αvβ3, αvβ5 and α5β1 integrins. Of all the integrins, αvβ3 appears to be the most important during tumor angiogenesis, as it is highly expressed in tumor vascular endothelial cells as well as some tumor cells, but not in most normal organs, making it a target for antiangiogenic therapy. When administered, integrin αvβ3 inhibitors, such as antibodies, peptides and other antagonists, bind to proteins on the surface of blood vessels and block the binding of αVβ3 integrin and its ligands, preventing the growth of new blood vessels and leading to the inhibition of angiogenesis and metastasis. The other is platelet glycoprotein (GP) IIb/IIIa receptor inhibitor (GPI). GPI inhibits platelet aggregation by binding to GP IIb/IIIa receptors, which acts at the last stage of platelet aggregation (Table 2).

### 6.1. Integrin Receptor Antagonists

SF0166, developed by Scifulor is a kind of effective small molecule inhibitor for αvβ3 integrin. As a kind of eye drops, SF0166 is used to treat eye diseases, such as AMD and DME. Preclinical data show that SF0166 is effective in an in vivo model of wet AMD [95].

Flotegatide (18F, 18F-SMIBR-K5, 18F RGD-K5) is based on galactosyl RGD that targets αvβ3 integrin. It is an injectable RGD peptide developed by Siemens Medical Solutions Molecular Imaging. 18F has been used for positron emission tomography (PET) imaging of tumor angiogenesis and clinically for the selective extraction of plaque in an ApoE knockout mouse model of atherosclerosis [96].

Thr-687 is an effective integrin antagonist, which can block the major RGD integrins αvβ3, αvβ5 and α5β1 to participate in the angiogenesis, inflammation, fibrosis and vascular permeability of pathological diseases. Thr-687 showed significant antiangiogenic effects in relevant in vitro, ex vivo and in vivo models [97].

### 6.2. Platelet Receptor Antagonists

Battifiban is a synthetic platelet glycoprotein IIb/IIIa receptor antagonist, which not only has a strong affinity for the target to inhibit platelet aggregation, but can also suppress the growth of vascular smooth muscle by inhibiting the VTN receptor so as to prevent arterial reocclusion [98].

Abciximab can inhibit the communication between vascular endothelium and the receptor αVβ3 integrin in smooth muscle cells [99]. However, abciximab is a human–mouse chimeric antibody. Although its immunogenicity is greatly reduced, it was found that abciximab still caused an anti-mouse antibody immune response in many patients and caused thrombocytopenia in clinical application [100].

Tirofiban is a synthetic GP IIb/IIIa receptor antagonist, which is a small molecule non-peptide RGD that shows highly specific binding to platelet GP IIb/IIIa receptors. It is a specific competitive inhibitor of the GP IIb/IIIa receptor [101].

## 7. Conclusions

The present study shows that the ECM does not merely act as an inert support as previously thought, but rather contains a large number of signaling molecules. As both a marker and nourishing factor in cell culture, VTN has become the concern in inflammatory, cancer and neurodegenerative diseases. It can change the state of neurons and affect the brain by mediating integrins. VTN is involved in various stages of neurons and glial cells: on the one hand, it can induce and enhance the differentiation of embryonic stem cells and oligodendrocytes, support guiding axon growth and repair damaged axons. On the other hand, VTN and its receptors can influence the phagocytosis of microglia and play a role in neurodegenerative diseases such as AD and AMD. VTN also affects the heterogeneity of the astrocyte response to promote the expression of CNTF, leading to neurogenesis and neuroprotection. Dysfunction and misfolding of VTN can lead to the pathology of neurodegenerative diseases and BBB leakage. The mechanism as to how VTN affects neuron activity may help to further clarify the pathogenesis of neurodegenerative diseases and facilitate the diagnosis and treatment of these diseases. In summary, there is increasing evidence that VTN exhibits more capacity for regulating neural function than previously thought, and future studies should focus on: (1) the mechanisms of VTN in neurodegenerative diseases; (2) the expression of VTN in the aging process and its effect on neurons; (3) investigating whether VTN-related drugs can ameliorate neuronal damage; (4) how to open the BBB through VTN to deliver drugs.

## Figures and Tables

**Figure 1 ijms-23-12387-f001:**
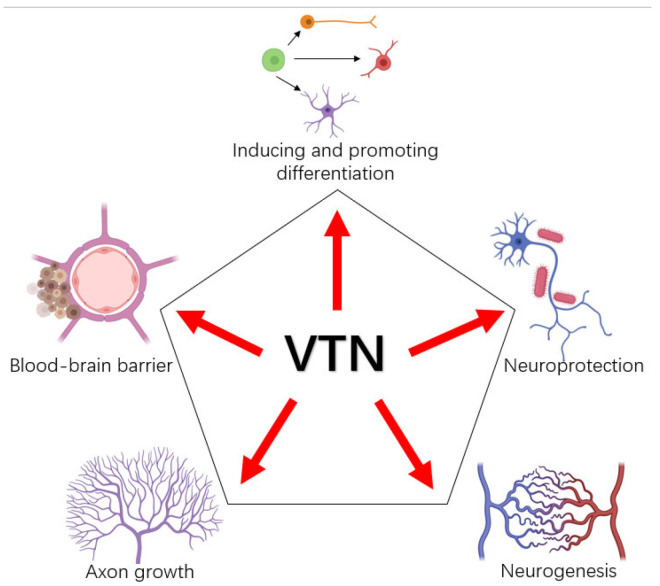
Overview of functions of VTN in neurons. Created with BioRender.com (accessed on 27 August 2022).

**Figure 2 ijms-23-12387-f002:**
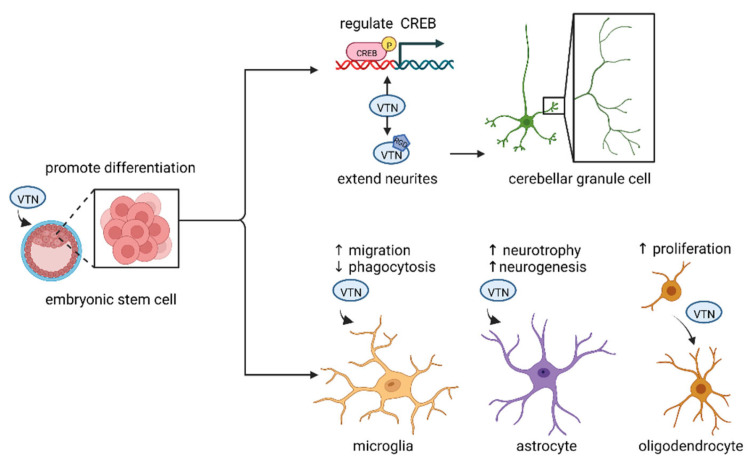
Schematic representation of VTN functions in different cells of the nervous system. The differentiation of embryonic stem cells can be promoted by changing ECM and selecting VTN. Furthermore, VTN acts on cerebellar granule cells by 2 ways: inducing CREB phosphorylation to regulate differentiation and utilizing RGD site to extend neurites, and it is also closely related to glial cells. VTN can promote microglia migration and inhibit phagocytosis. Moreover, it can induce neurogenesis and neuroprotection of astrocytes and proliferation of dendritic cells. Created with BioRender.com (accessed on 11 October 2022).

**Figure 3 ijms-23-12387-f003:**
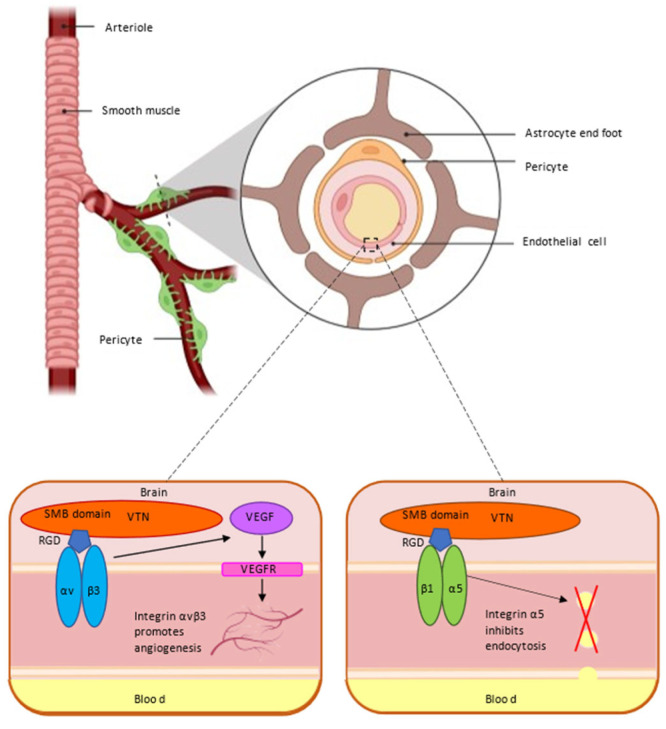
A schematic illustration depicting how VTN and its receptors αvβ3 and α5β1 affect blood–brain barrier. On the left, VTN by binding to αvβ3 can activate VEGF, which regulates angiogenesis by binding to VEGFR on the surface of cell membranes. On the right, integrin α5 can inhibit endocytosis of endothelial cells to ensure the integrity of barrier function and protect the brain. Adapted from “Brain Vascular System”, by BioRender.com (2022). Retrieved from https://app.biorender.com/biorender-templates (accessed on 27 August 2022).

**Table 2 ijms-23-12387-t002:** Current targeted drugs for VTN and its receptors

Name	Function	Efficacy	Company
SF0166	αvβ3 integrin inhibitor	effect	Scifulor
Flotegatide	RGD peptide	effect	Siemens Medical Solutions
Thr-687	integrin antagonist	ongoing	Oxurion
Battifiban	platelet GPIIb/IIIa receptor antagonist	effect	DongRui (Hangzhou) Medical Technology
Abciximab	platelet GPIIb/IIIa receptor antagonist	effect	Janssen Biologics BV
Tirofiban	platelet GPIIb/IIIa receptor antagonist	effect	Medicure Pharma

## Data Availability

Not applicable.

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
