# Peer review of "Role of Vitronectin and Its Receptors in Neuronal Function and Neurodegenerative Diseases"

_ijms, 2022, doi:10.3390/ijms232012387_

Round 1
Reviewer 1 Report
VTN has attracted atention as the ECM has been poor studied at the CNS. This review shows the complex role of this protein both during development and neurodegeneration (ND).
There is a pair of consideration :
1. Please, separate each subsection from the above one.
2. It would be recommendable to summarize information about VTN and ND and VTN and development in graphs or table as the authors performed with the BBB
Author Response
VTN has attracted atention as the ECM has been poor studied at the CNS. This review shows the complex role of this protein both during development and neurodegeneration (ND).
There is a pair of consideration :
- Please, separate each subsection from the above one.
[Author’s response]
Thank you for this valuable suggestion. We have separated each subsection from the above one in the revised manuscript.
- It would be recommendable to summarize information about VTN and ND and VTN and development in graphs or table as the authors performed with the BBB.
[Author’s response]
We appreciate the reviewer for the pertinent comment. We have summarized the functions of VTN in neurodegenerative diseases and in different cells of the nervous system, which was presented as graph or table in the revised manuscript, please refer to Figure 2 and Table 1.
Reviewer 2 Report
This is a well-written review manuscript that comprehensively analyze the role of vitronectin in neuronal function and neurodegenerative diseases. The illustrations are good quality.
Author Response
This is a well-written review manuscript that comprehensively analyze the role of vitronectin in neuronal function and neurodegenerative diseases. The illustrations are good quality.
[Author’s response]
Thank you for your approval of this manuscript.